# Influence of COVID-19 Pandemic on Colorectal Cancer Presentation, Management and Outcome during the COVID-19 Pandemic

**DOI:** 10.3390/jcm12041425

**Published:** 2023-02-10

**Authors:** B. M. Pirozzi, L. Siragusa, G. Baldini, M. Pellicciaro, M. Grande, C. Efrati, R. Finizio, V. Formica, G. Del Vecchio Blanco, G. S. Sica

**Affiliations:** 1Department of Surgery, University of Rome “Tor Vergata”, 00133 Rome, Italy; 2Department of Emergency, University of Rome “Tor Vergata”, 00133 Rome, Italy; 3Department of Medicine, Israelitic Hospital of Rome, 00148 Rome, Italy; 4Department of Medicine, University of Rome “Tor Vergata”, 00133 Rome, Italy

**Keywords:** COVID-19, colorectal, cancer, pandemic

## Abstract

The aim of the study was to investigate whether the COVID-19 pandemic and related measures had an influence on colorectal cancer (CRC) presentation, management, and outcomes; it was a retrospective monocentric study. CRC patients undergoing surgery during the COVID-19 pandemic (1 March 2020–28 February 2022) (group B) were compared with patients operated on in the previous two years (1 March 2018–29 February 2020) in the same unit (group A). The primary outcome was to investigate whether there were differences in concern regarding the stage at presentation, as a whole and after dividing groups based on cancer location (right colon cancer, left colon cancer, rectal cancer). Secondary outcomes included differences in the number of patients admitted from emergency departments and emergency surgeries between periods, and differences in the postoperative outcomes. A subanalysis within the pandemic group was conducted on the same outcomes, dividing the aforementioned group based on pandemic trends. Two hundred and eighty (280) were operated on during the study period: 147 in group A and 133 in group B. Stage at presentation was similar between groups; however, the subgroups analysis showed that in the pandemic group, the number of early-stage left colon cancer occurrences almost halves, yet not significantly. Emergency department referral was more common in group B (*p*-value: 0.003); in group B, they also had longer operations and there was a more frequent use of ostomy. No differences in the number of postoperative complications nor in the postoperative outcomes were found. Patients with CRC were more frequently referred through the emergency department during the COVID-19 pandemic and left-sided cancers appear to be generally diagnosed at a more advanced stage. Postoperative outcomes showed that high specialized colorectal units can deliver standard high-level treatment under high-pressure external conditions.

## 1. Introduction

Significant Acute Respiratory Syndrome, named coronavirus-2 (SARS-CoV-2), is a novel coronavirus with an infectious etiology contracted through zoonotic transmission. In the Chinese city of Wuhan, the first clinical case occurred on 8 December 2019, but the novel coronavirus was not discovered until 7 January 2020, and it spread rapidly before the World Health Organization declared a global alert level [1]. Italy was placed under lockdown on 10 March 2020, as a result of the virus propagation having touched the Italian peninsula, particularly the north, as of February 2020 [2]. The Italian National Health System was compelled to rearrange its financial resources and staff due to the pandemic emergency. Various activities, including screening for colorectal cancer (CRC), have been discontinued and elective surgery has been drastically scaled back [3].

The effect of COVID-19 on CRC screening, diagnosis and treatments has depended not only on the allocation of Health System resources, but also on people seeking less medical care, including those with acute or chronic health conditions in follow-up, and those undergoing other treatments. This behavior could be related to the fear of getting the infection while receiving hospital treatment, or else it could be explained by the desire to avoid further overburdening the healthcare system, already stressed by the pandemic emergency [4,5,6].

According to Globocan, CRC is the third most common cancer and the second by mortality [7]. Despite these statistics being maintained in 2020, numerous studies predict a decrease in the number of new instances of CRC being diagnosed. Morris et al. [8] calculated that around 3500 fewer people in England were diagnosed and treated for CRC than estimated (based on the statistics from the previous year); the same trend was also seen in other countries, including the Netherlands [9], Brazil (where a 46% reduction in new diagnoses was calculated) [10], and Canada, where the calculated reduction was of 48%.

The COVID-19 pandemic has transformed medical care worldwide. General surgery has been affected in elective and emergency procedures. The 2021 Italian National Healthcare Outcomes Program (PNE) calculated the hospitalization volume (Figure 1) for colorectal cancer surgery, showing a decrease both for colon (23,078 in 2020 vs. 26,233 in 2019) and rectal cancer (5627 vs. 6051) [11].

A multi-center study from Germany showed that the 2020 lockdown significantly impacted on abdominal emergencies and led to a 13% reduction in appendectomies. While the absolute number decreased, the rate of complicated appendicitis increased significantly, possibly due to a longer waiting time between the onset of symptoms and medical consultation [12,13,14,15,16,17].

Herein is reported a retrospective analysis from a single unit, in order to assess the short- and medium-term effects of the pandemic on the clinical onset of CRC. The goal was to determine if the parameter—and eventually which parameter—was affected by the containment measures and reallocation of healthcare resources. In particular, the principal aim was to evaluate whether it led to a delayed diagnosis and, consequently, to a higher disease stage and complications such as bowel occlusion and perforation, eventually affecting surgical treatments and short-term outcomes.

## 2. Methods

### 2.1. Inclusion and Exclusion Criteria

All patients with CRC undergoing colorectal resection at the Department of Surgery, Minimally Invasive Unit, University Hospital Tor Vergata in Rome, Italy, between March 2018 and February 2022, were eligible for the study and evaluated for inclusion.

The only exclusion criterion was the COVID-19 positivity during the swab test.

Ethical committee approval according to institutional evaluation was deemed not necessary, given the retrospective design of the study and the analysis of the anonymized data.

### 2.2. Study Design

It was a retrospective, comparative study from a single unit. Data were retrieved from a prospectively maintained database using anonymized patients’ data, to compare CRC presentation and staging according to the 1987 TNM Classification of Malignant Tumors (TNM) [18] and the treatment and outcomes in patients undergoing surgery before and during the COVID-19 pandemic.

The study adhered to the Strengthening the Reporting of Observational Studies in Epidemiology (STROBE) Statement.  

Patients undergoing surgical resection during the COVID-19 pandemic (1 March 2020–28 February 2022, group B) were compared to a similar cohort of patients who had had surgery in the previous two years (1 March 2018–29 February 2020, group A). To reduce biases related to evolving surgical practices, expertise, and incidence rates, patients for the control group were those undergoing surgery in the two years just before the pandemic.

A first subanalysis was conducted by dividing the two cohorts based on cancer location (right colon cancer, left colon cancer, and rectal cancer); a second subanalysis was conducted by dividing group B on time-based pandemic trends (peak COVID-19, post peak COVID-19). We referred to the Italian Government monitoring report [2] in order to identify the peak-pandemic phases (March–May 2020; October 2020–January 2021; March–June 2021) when the lockdown measures were stricter and healthcare access was limited vs. the post-peak phases (June–October 2020; February–March 2021; July 2021–February 2022) when limitations were softened.

The preoperative evaluation included the patient’s clinical data, such as age, sex, weight, height, BMI, ASA Physical Status Classification System (as per the American Anesthesiology Association [19]), signs and symptoms before cancer diagnosis, and regimen of hospital admission. Concerning peri-operative data, it was recorded whether the procedure was carried out by an attending surgeon or a surgeon in training; also recorded were the surgical approach (open, laparoscopic, conversion), the operative time, and the need for an ostomy. Surgeries were performed following oncological principles if not contraindicated. Patients with right colon cancer were randomly assigned to complete a mesocolic excision as part of an undergoing study [20]. Peri-operative care followed the ERAS pathway, when applicable, for the whole study period [21].

### 2.3. Outcome Measures

Primary outcome: to find the differences in stage at presentation between groups as a whole and after subdividing cohort into three subgroups according to cancer location: right colon, left colon, and rectum.

Secondary outcomes: to find the differences in the number of patients referred from the Emergency Department, numbers of urgent procedures, and postoperative outcomes such as duration of hospital stay, applicability of the ERAS pathway, number of CT scans requested, number of major postoperative complications, defined as the Clavien–Dindo category ≥ 3 [22], histopathological presentation, and mortality defined as any cause of death within 30 days from intervention.

Additionally, a further subanalysis of the pandemic group was conducted in order to investigate differences in cancer presentation based on pandemic trends (peak COVID-19, post-peak COVID-19).

### 2.4. Statistical Analysis

Characteristics were summarized by means of the levels for categorical variables or by means of the quantiles for the continuous variables. Non-parametric tests were performed for comparisons between groups (the chi-squared and Fisher Exact tests in case of categorical variables, the Wilcoxon test in case of continuous variables). The Cox–Stuart test was used to test whether the data have an increasing or decreasing trend. All tests were 2-sided, accepting *p* < 0.05 as indicating a statistically significant difference and confidence intervals were calculated at a 95% level.

All the statistical analysis was performed in SPSS statistical package version 23.0 (SPSS Inc., Chicago, IL, USA).

## 3. Results

A total of 280 patients undergoing surgery for CRC were included: 133 patients (47%) underwent surgery during the SARS-CoV-2 pandemic (Group B) and 147 (52%) underwent surgery before the pandemic (Group A). All patients were negative for COVID-19 infection, detected using a PCR test on nasal and oropharyngeal swabs, before surgery and on the day of discharge.

In Table 1 are reported the differences amongst the two groups in the pre-operative variables.

The two groups were homogeneous for most preoperative characteristics but in group B there were more ASA 4 patients (8% in group B vs. 3% in group A) and significantly less ASA 1 patients (2% in group B vs. 11% in group A; *p*-value = 0.006). The number of right colon cancer was higher in group B, although not significantly (46% in group B vs. 39% in group A) while the proportion of left and rectal cancer remained almost the same in the two groups (33% in group A vs. 29% in group B and 28% in group A vs. 25% in group B, respectively). The regimen of surgery did not vary significantly in the two groups (elective surgery: 84% group A vs. 92% group B; emergency surgery: 16% group A vs. 8% group B). There was an increased numbers of patients referred from the emergency department during the pandemic (49% group A vs. 67% group B; *p*-value: 0.003); laparoscopic resections remained stable (63% group A vs. 68% group B); the operative time during the pandemic was higher (164 min group A vs. 191 min group B; *p*-value 0.0001) like the numbers for an ostomy (16% group A vs. 29% group B; *p*-value: 0.013); the requests for a postoperative CT scan to rule out complications showed a slight increase (24% group A vs. 29% group B). Concerning the number of operations undertaken by surgeons in training, there was a reduction during the pandemic (6% in group B vs. 16% in group A; *p*-value: 0.011), whilst the number of patients undergoing ERAS protocol was similar between groups (62% group A vs. 63% group B).

The same variables were all analyzed again according to time, specifically group B, which was subdivided into the peak COVID-19 group (group C) and post peak COVID-19 (group D). Preoperative parameters (Age, BMI, ASA score, cancer location, and surgical regimen) did not show a significant difference between the two subgroups with the exception for the sex distribution (46% male in group C vs. 68% male in group D; *p*-value 0.01), pre-operative hemoglobin (12.3 ± 2.2 g/dL in group C vs. 11.4 ± 2.1 g/dL in group D; *p*-value 0.017), and emergency department referral (56% in group C vs. 76% in group D: *p*-value 0.025).

Concerning the primary aim, the tumor stages did not show significative differences, as shown in Table 2.

An analysis of the subgroups also could not find differences in the tumor stages when the two groups were divided according to location or according to time (Table 3). Nevertheless, for left side tumors, the amount of stage I cancers almost halves during the pandemic and doubles for stage II.

When looking at the histopathological data, the two groups did not report significant differences in the nodal distribution, EMVI, tumor grading, histotype, or grading. There was instead a statistically significant difference in the metastasis location between the two groups in terms of the general cohort (liver: 2% group A vs. 8% in group B; *p*-value: 0.044; peritoneum: 1% in group A vs. 2% in group B; lung: 2% in group A vs. 4% in group B) but not in the subdivision of the pandemic group during peak COVID-19 vs. post peak COVID-19, as reported in Table 4.

Postoperative outcomes are reported in Table 5. The complication rate (Clavien–Dindo ≥ 3 group A 11% vs. group B 16%) and 30 days mortality rate (1% group A vs. 2% group B) did not vary significantly during or before the pandemic. Particularly, there were 1% of the anastomotic leaks in group A vs. 2% of the anastomotic leaks in group B; 5% of patients in group A underwent reintervention vs. 4% of patients in group B, while there were 11% SSI in group A vs. 5% in group B. The median hospital stay was identical in the two groups (5 days).

Other variables with statistical significance, such as the ASA score, the number of ostomies, the operative time, and emergency department presentation, are reported in Figure 2.

## 4. Discussion

The COVID-19 pandemic demonstrated the fragility of oncologic surgery setups worldwide. An international study by Glasbey et al., spanning 20,006 solid organ cancer patients across 61 countries showed that, in regions with full lockdowns, one in seven patients did not undergo planned surgery and experienced longer preoperative waiting times. Findings such as an increase in the positive resection margins or new metastatic disease due to the delays in the treatments highlight compromising short and middle term outcomes; furthermore, delays and missing operations certainly might affect long-term outcomes including disease-free survival [23,24].

There are little data regarding the clinical presentation and surgical outcomes of CRC during the pandemic. The present study focuses on CRC patients undergoing surgery in the period that goes from March 2020 to February 2022, when the government activated different levels of the public health emergency response. The principal aim was to investigate the impact of the pandemic on the diagnosis, treatment, and short-term outcome of CRC patients undergoing surgical resection. As part of the strategy of prevention, with the exception of the first three months of the pandemic, in which most of the services were shut and only the emergencies were taken care of, in-hospital routine medical services were continuously adjusted based on the changes and requirements for the prevention and control of the pandemic. Most of the benign pathology commonly treated in colorectal units were discontinued [25,26,27]. Nevertheless, even during the lockdown, oncology patients were assured priority for receiving a standard of care treatments. Thanks to team efforts, oncological units were capable to keep the latency between CRC diagnosis and treatment below the 6-week range, as required by ESMO guidelines [28]. As such, we reported the same experience by not prolonging the interval between diagnosis and surgery for the time period analyzed over a mean time of 15 days. However, several reports, from Spain, Japan, and other countries [29,30,31,32] described a generalized increase in stage III and IV tumors during the pandemic. We could not find any differences in stage at presentation nor in histopathological data between groups (Table 2, Table 3 and Table 4), but the subgroup analysis showed, only for left colon tumors, an increase of almost double for stage II cancers and a drop in stage I. This is possibly explained by the interruption of the CRC screening program [33,34,35,36,37,38,39,40,41]. Many studies demonstrated that advanced left-sided cancers (including rectal cancer) decreased after the endoscopic screening protocol introduction while right-sided cancers have a lower decrease [42,43]. He Yong et al. simulated the detection and progression of CRC using OncoSim (version 3.3.6), describing the trends towards an increase in adenocarcinomas findings because of a major number of undiagnosed adenomas [41]. A research model from the UK predicted that delays in diagnosis brought about by the COVID-19 pandemic would shorten the long-term survival time of CRC patients, potentially causing a 15–17% increase in mortality within 5 years of diagnosis [44]. Therefore, even if the gross number of cases treated during the pandemic has remained about the same, and even if this study is biased by its small numbers, there are probably several undiagnosed tumors, especially early-stage cancers, which will manifest more aggressively in the near future [45]. If this prevision is correct, this situation will affect the long-term prognosis with an increase in the 5-year mortality [31]. Complicated cases are to be foreseen, as it has been shown that a diagnostic delay of more than 4 months correlates with an increased risk of incurring an in-bowel obstruction as an onset symptom. At the same time, emergency treatments increase the risk of postoperative complications, with an increased “loss of life years” [37,46,47,48]. Furthermore, Morris et al. have observed that CRC diagnosed at a more advanced stage with severe symptomatology (anemia, obstruction, intestinal perforation) have a negative impact on prognosis [8].

To this day, we have already found that a sizable portion of patients have been diagnosed with CRC after accessing the emergency department because of a symptomatology no longer sustainable. As listed in the study by D’Ovidio et al. [40], this increase was mostly related to the burden of the pandemic on general practitioner and specialists in the territory. Evidence suggests that the lockdown led to a 30% reduction in primary care consultations [49].

However, it seems that these figures were not the same everywhere; some institutions demonstrated the capacity to guarantee the necessary therapies in a timely manner, showcasing the various ways that lockdown has hindered the effectiveness of global health services while considering the facilities’ ability to manage an emergency of a similar phenomenon as a separate variable [36,50]. We observed an increase in access through the emergency department, but eventually it was possible to delay surgery so that there was not a significative increase in emergency procedures during the pandemic. Every patient requiring an emergency procedure for severe symptoms such as perforation or severe occlusion were treated by the department of emergency and for this reason were excluded from this study; on the other hand, every patient that did not require an emergency procedure (defined as surgery performed within 12 h from admission) were stabilized and sent to our Colorectal Unit and, when possible, there was a delay to the emergency for further assessment and treatment. Patients in group B presented with more co-morbidities and the numbers of ASA 4 patients was higher, whilst, on the contrary, ASA 1 patients were absent. In the subset of analysis related to the pandemic trends, there was an increase in male gender among the post peak COVID-19 group, a lower hemoglobin level and a higher access through the emergency department, as shown in Table 6. These differences are most likely related to the way of presentation of the two groups of patients (group A vs. group B), with the pre-COVID-19 patients being referred to surgical wards as a consequence of screening programs in many instances. Whereas specifically looking at group B, during the peak pandemic patients would not refer to medical attention unless it was utterly necessary, maybe because some were fearful of contracting COVID-19 from hospitals [4,5], while others were afraid to overburden the healthcare systems [35].

Concerning the surgical approach, there were major changes worldwide in the delivery of treatment. The number of operations fell sharply, and surgical methods were adapted to minimize COVID-19 risk [45]. In Italy, the PNE detected a decrease in laparoscopic surgeries in 2020 compared to 2019 (10,754 vs. 12,515 for colon and 3075 vs. 3261 for rectum) [11].

The present study shows no differences in the number of laparoscopic colorectal resection during the pandemic. However, there was indeed an increase in ostomies as reported by other centers worldwide [51,52,53]. The higher stoma rate is mostly associated to the patient-related risk factors for anastomotic leakage according to Spinelli et al. [54]. There were a higher number of ASA 4 in group B vs. group A and less ASA 1 in group B vs. group A; also, in group B there were more patients with cardiovascular disease, males, smokers, and lower haemoglobin, all factors that, according to the literature, would expose the patient themselves to an increased risk of developing postoperative complications. This, together with the EAGLE guidelines [55] and the international recommendation of increasing the in-hospital flow during the pandemic [56], led our unit to protect more anastomosis than the previous time frame, in order to maintain the risk of reoperation at its minimum, by increasing in-hospital patient flow and minimizing in-hospital infections, reducing the risk of nosocomial COVID-19 infection in patients in a fragile condition. Possibly for the same reason, there were less operations performed by surgeons in training and an increased number of post-operative CT scans to rule out potential post-operative complications. The choice to use the CT scan more frequently during the COVID-19 period was related to the higher concern about post-operative complications, not only to exclude as soon as possible surgical complications that were the main indications for CT execution, but also to reduce hospital stay to its minimum by testing borderline patients that otherwise would probably have a longer mean hospital stay. Normally at our institution, the indication for a CT scan is given with, on the third postoperative day, a PCR above 150 mg/L or three times higher than the one seen on the first postoperative day, according to Spinelli et al. [54]. During the pandemic, though, we lowered that threshold to 100 mg/L, in order to exclude potential surgical complications and to discharge patients earlier on with the intention of reducing the hospital stay and to increase in-hospital flow again to minimize the risk of in-hospital infection.

As a matter of fact, the mean postoperative hospital stay in Italy after colorectal resection for cancer decreased slightly (median was 8 days in 2019 vs. 7 days in 2020; Figure 3) [11]. Concerning the peri-operative care, there are reports showing that the ERAS pathway, due to its complexity, cannot be carried out in a period of health crisis [57]. We have already described our experience with ERAS and gastrointestinal surgery during the pandemic [21]. We found the same adherence to the ERAS pathway during the two-study period and similar length of hospital stay (median 5 days). The fairly high rate (63%) of ERAS adherence reflects the great effort of a highly specialized unit, to deliver the same standard of care also in a situation of high pressure. The PNE registered an increase in the mortality rate after surgery for colon cancer (3.89% in 2019 vs. 4.83% in 2020), most likely related to patients concurrently affected by COVID-19 infections that have not been excluded from the general PNE survey [11]. Mortality in the present study did not differ within the two groups (group A 1% vs. group B 2%). No death was related to COVID-19 since the only exclusion criteria was COVID-19 positivity during the PCR test for SARS-CoV-2. Every patient in this study tested negative for COVID-19 during the swab test 48 h before surgery and at discharge. As for post-operative outcomes and complications, the number of patients with Clavien–Dindo scores greater than three has increased during the pandemic, yet not reached statistical significance and could be due to the increased number of patients with severe comorbidities [58].

The present study shows no particular differences in the primary nor secondary outcome, even when it comes to a specific subset of analysis in which group B was further divided into two subgroups according to the pandemic waves in Italy when the lockdowns were stricter and health services were delayed [2]. The COVID-19 group was divided into peak COVID-19 (group C) when health services were discontinued and post-peak COVID-19 (group D) when limitations were softened. As shown in Table 6, the only data worth mentioning are the different distributions in sex and lower hemoglobin for the post peak COVID-19 group (group D) with an increased access through the emergency department. This data helps confirm our assumption that more stomas are created during the COVID-19 period compared to the control group (group A), respecting the EAGLE guidelines [56] and the Italian consensus on management of anastomotic leakage [55].

The present study has severe limitations due to the retrospective nature of the analysis and the small sample size. Nevertheless, a prospective study is undoubtedly difficult if ever possible. Such information would have greater value if a multicenter study was used. However, in this case, we chose to conduct a single-center study in order to share our internal data on the specific issue at hand. The benefits of this study could be related to demonstrating that specialized centers are able to provide high-quality care even under severe external stress and was developed with the only aim of improving clinical practice, if ever again under similar circumstances. The two groups were homogeneous and the differences amongst the groups were observed and discussed after setting the aims of the study so as to adhere to the STROBE statement.

## 5. Conclusions

No major differences were found in patients with CRC during the COVID-19 pandemic comparing them with a pre-pandemic population. The subanalysis via cancer location (right, left or rectal cancer) or via time (peak COVID-19 vs. post peak COVID-19) did not show significant differences. Patients were more compromised and more often were referred from the emergency department. Postoperative outcomes from a specialized colorectal unit, following ERAS protocol, were similar to the pre-pandemic period, demonstrating to be able to deliver high quality, optimal treatment even under high-pressure external conditions.

## Figures and Tables

**Figure 1 jcm-12-01425-f001:**
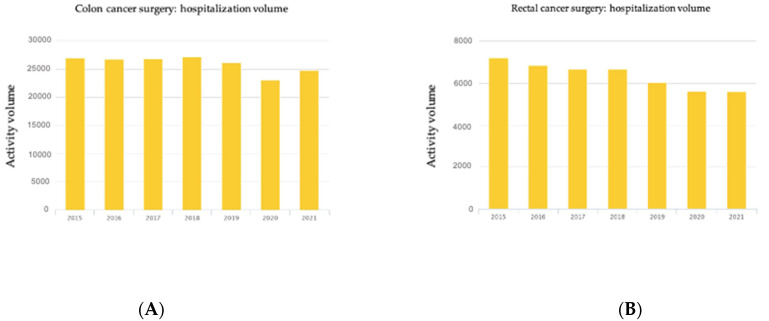
Hospitalization volume for colon cancer (**A**) and rectal cancer (**B**) in Italy according to PNE 2021 [11].

**Figure 2 jcm-12-01425-f002:**
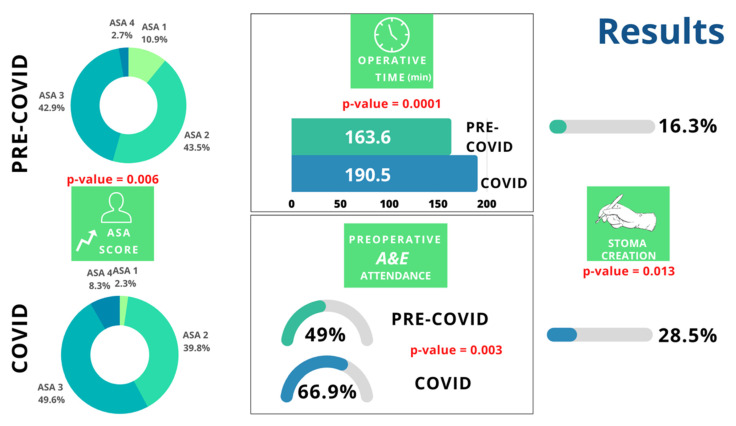
Analysis of statistically significant variables.

**Figure 3 jcm-12-01425-f003:**
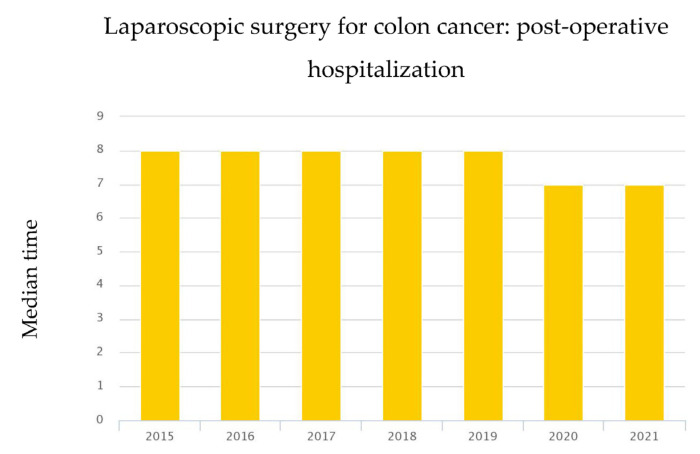
Post-operative hospitalization (median in days) after surgery for colon cancer in Italy according to PNE [11].

**Table 1 jcm-12-01425-t001:** Preoperative parameters.

Parameters		Group A (n = 147)		Group B (n = 133)		*p*-Value
Age (mean, SD) Year		69.5 ± 12.8 Median 72		70.7 ± 11.5 Median 74		0.452
Sex	Male	84	57.1%	77	57.9%	0.904
(%)	Female	63	42.9%	56	42.1%	
BMI (mean, SD)		25.3 ± 4.3 Median 25		26.2 ± 4.0 Median 25.8		0.072
ASA score %						0.006
1		16	10.9%	3	2.3%	
2		64	43.5%	53	39.8%	
3		63	42.9%	66	49.6%	
4		4	2.7%	11	8.3%	
Localization						0.534
Right Colon		57	38.7%	61	45.9%	
Left Colon		49	33.3%	38	28.6%	
Rectum		41	27.8%	34	25.5%	
Time from diagnosis to Surgery		12 ± 3.3 Median 14		13 ± 2.8 Median 15		0.962

**Table 2 jcm-12-01425-t002:** Primary Outcomes.

Staging	Group A (n = 147)		Group B (n = 133)		*p*-Value
I	22	15%	24	18%	0.552
II	46	31.3%	48	36.1%	
III	56	38.1%	46	34.6%	
IV	23	15.6%	46	34.6%	
**Staging** **Right Colon Cancer**	**Group A (n = 57)**		**Group B (n = 61)**		***p*-value**
I	5	8.79%	10	16.4%	0.620
II	22	38.5%	20	32.8%	
III	21	36.8%	23	37.7%	
IV	9	15.8%	8	13.1%	
**Staging** **Left Colon Cancer**	**Group A (n = 49)**		**Group B (n = 38)**		***p*-value**
I	11	22.4%	5	13.1%	0.150
II	12	24.5%	18	47.4%	
III	21	42.9%	13	34.2%	
IV	5	10.2%	2	5.3%	
**Staging** **Rectal Cancer**	**Group A (n = 41)**		**Group B (n = 34)**		***p*-value**
I	6	14.6%	9	26.5%	0.606
II	13	31.7%	10	29.4%	
III	13	32.5%	10	29.4%	
IV	9	21.9%	5	14.7%	

**Table 3 jcm-12-01425-t003:** Primary outcomes for pandemic subset of analysis.

Staging	Group C (n = 59)		Group D (n = 74)		*p*-Value
I	13	22%	11	14.9%	0.685
II	21	35.6%	27	36.5%	
III	18	30.5%	28	37.8%	
IV	7	11.9%	8	10.8%	
**Staging** **Right Colon Cancer**	**Group C (n = 24)**		**Group D (n = 37)**		***p*-value**
I	7	29.2%	3	8.1%	0.103
II	7	29.2%	13	35.1%	
III	6	25%	17	46%	
IV	4	16.6%	4	10.8%	
**Staging** **Left Colon Cancer**	**Group C (n = 16)**		**Group D (n = 22)**		***p*-value**
I	2	12.4%	3	13.6%	0.973
II	7	43.8%	11	50%	
III	6	37.5%	7	31.8%	
IV	1	6.3%	1	4.6%	
**Staging** **Rectal Cancer**	**Group C (n = 19)**		**Group D (n = 15)**		***p*-value**
I	4	21%	5	33.3%	0.601
II	7	36.8%	3	20%	
III	6	31.7%	4	26.7%	
IV	2	10.5%	3	20%	

**Table 4 jcm-12-01425-t004:** Histopathological data.

Histopathological Data	Group A (n = 147)		Group B (n = 133)		*p*-Value
N+ %	74	50.3%	55	41.4%	0.150
EMVI	64	43.5%	64	48.1%	0.472
METS SITE					
Liver	3	2%	10	7.5%	0.044
Peritoneum	1	0.6%	3	2%	0.394
Lung	3	2%	5	3.6%	0.484
Grade					0.624
1	12	8%	8	6%	
2	84	57.1%	61	45.8%	
3	51	34.6%	64	48.1%	
HISTOLOGY					0.261
Adenocarcinoma	104	70.7%	88	66.2%	
Mucinous	39	26.5%	36	27.1%	
Signet-cell	2	1.4%	7	5.3%	
Other	2	1.4%	2	1.5%	
**Histopathological** **Data**	**Group C (n = 59)**		**Group D (n = 74)**		***p*-value**
N+	23	39%	32	43.2%	0.732
EMVI	31	52.5%	33	44.6%	0.387
METS SITE					
Liver	4	6.8%	6	8.1%	1
Peritoneum	1	1.6%	2	2.7%	1
Lung	2	3.4%	3	4.1%	1
Grade					0.384
1	3	5.0%	5	6.7%	
2	31	52.5%	30	40.5%	
3	25	42.3%	39	52.7%	
HISTOLOGY					0.372
Adenocarcinoma	37	62.7%	51	68.9%	
Mucinous	15	25.4%	21	28.3%	
Signet-cell	6	10.2%	1	1.4%	
Other	1	1.7%	1	1.4%	

**Table 5 jcm-12-01425-t005:** Secondary outcomes.

Parameters	Group A (n = 147)		Group B (n = 133)		*p*-Value
Emergency Department Attendance	72	49%	89	66.9%	0.003
Regimen of Surgery					0.068
Elective	124	84.4%	122	91.7%	
Emergency	23	15.6%	11	8.3%	
Hospital stay Days	6.3 ± 3.7 Median 5		6.2 ± 4.6 Median 5		0.841
ERAS compliance	92	62.5%	84	63.1%	0.905
Clavien-Dindo > 3	16	8.8%	21	15.7%	0.223
30 days mortality	2	1.4%	3	2.3%	0.671

**Table 6 jcm-12-01425-t006:** Pre-operative and post-operative parameters for pandemic subset of analysis.

Parameters		Group C (n = 59)		Group D (n = 74)		*p*-Value
Age (mean, SD) Year		69.3 ± 12.1 Median 73		71.6 ± 11 Median 74		0.252
Sex	Male	27	45.8%	50	67.6%	0.012
(%)	Female	32	54.2%	24	32.4%	
BMI (mean, SD)		26.3 ± 4.3 Median 26		26.1 ± 3.8 Median 26		0.776
Localization						0.282
Right Colon		24	40.7%	37	50%	
Left Colon		16	27.1%	22	29.7%	
Rectum		19	32.2%	15	20.3%	
Hemoglobin (gr/dl) (mean, SD)		12.3 ± 2.2		11.4 ± 2.1		0.017
Regimen of Surgery						0.754
Elective		55	93.2%	67	90.5%	
Emergency		4	6.8%	7	9.5%	
Emergency Department Attendance		33	55.9%	56	75.7%	0.025
Stoma creation		16	27.1%	17	22.9%	0.106
ERAS compliance		40	67.8%	46	62.1%	0.196
Hospital stay Days		6 ± 4.5 Median 5		7.4 ± 5.8 Median 6		0.13
Clavien-Dindo > 3		8	13.5	13	17.5	0.34
30 days mortality		2	3.4%	1	1.4%	0.587

## Data Availability

The data presented in this study are available on request from the corresponding author. The data are not publicly available due to privacy restriction.

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
