# Peer review of "Influence of COVID-19 Pandemic on Colorectal Cancer Presentation, Management and Outcome during the COVID-19 Pandemic"

_jcm, 2023, doi:10.3390/jcm12041425_

Round 1
Reviewer 1 Report
The study by Pirozzi et al. is well written and concise
It compares pre and post Covid impact on CRC surgery, outcomes, presentations etc.
I have few suggestions to help improve:
1. Would you consider dividing Covid into two subgroups - Peak Covid and then post peak Covid (when Italy started opening again to "normal" services)
We have saw a higher incidences of late presentations and advanced disease in the post peak Covid cohort
2. Have you any data on nodal differences between the time period, details on EMVI, tumor budding, other histopath and genetic details (MMR etc)
3. Any data on distribution of Stage IV - Liver, Lung or peritoneal
Again we have seen more peritoneal disease in post peak Covid
4. More details on why higher stoma rates in Covid group - I can guess why - But should be more specific
5. What were the main reasons for increased CT use in post-op period - What was main indications ? Was this surgical or Resp orientated ??
6. How many deaths were directly related to Covid-19?
7. Can you expand details on major morbidity (Clavien Dindo 3+)
8. Graphs have Italian - please amend
Author Response
Dear Reviewer,
please see the attachment.
Kind Regards
B.M. Pirozzi

Reviewer 2 Report
The information would have a greater value if a multicenter study was used.The benefits of this study must be clearly stated. The statistical method by which p-indexes were calculated must be clearly specified.
Author Response

(The authors gave the same response as above.)

Reviewer 3 Report
Good paper, decently written on an interesting topic for these periods. Few spelling issues to improve reading fluency.
Author Response

(The authors gave the same response as above.)

Reviewer 4 Report
The authors of this manuscript should be congratulated for their effort and for the general good quality of the manuscript. However, some minor comments from my side are listed herebelow:
- there are few minor grammar mistakes in the abstract that should be addressed. Furthermore, spell out numbers at the start of a sentence.
- in the abstract you mentioned that a subanalysis was performed with regard to three subgroups based on location. I would specify based on location of the tumor and would also mention the three subgroups (right, left, rectum cancers)
- results: since the majority of the results reported in the text are also reported in the tables i would be more concise about preop and staging characteristics. Results about postop outcomes defintely deserve more consideration in the results section.
- the results and discussion on the potential relation between the decrease of early stage left colon cancers --> screening --> potential burden of advanced cancers in the next few years is quite speculative. Merely statistically there is no difference detected in cancer stage before/during pandemic even for left side cancer. Secondly, the manuscritpt can not really prove that the decrease of early left cancers is due to the interruption of screening process during the pandemics (you have no data on this for patients included in this study). For this reason, i would be also quite prudent and would remove the last sentence of your conclusion.
- It seems like during the pandemics more elective and less urgent operations were performed (p is quasi significant)? This is quite paradoxical and difficult to explain in the setting of a pandemics. What is definition of urgent procedure in this study? (surgery performed within 24h from the admission?)
- do you have any data on the interval between diagnosis and surgery for both groups (since you mentioned this (from other studies) in your discussion)?
Author Response

(The authors gave the same response as above.)

Round 2
Reviewer 2 Report
After the changes made, from my point of view, the article is well written.